# The Regulatory Effects of JAK2/STAT3 on Spermatogenesis and the Redox Keap1/Nrf2 Axis in an Animal Model of Testicular Ischemia Reperfusion Injury

**DOI:** 10.3390/cells12182292

**Published:** 2023-09-16

**Authors:** Abdullah Alnajem, May Al-Maghrebi

**Affiliations:** 1Department of Medicine, Jaber Al-Ahmed Hospital, South Surra 47761, Kuwait; abdullah.s.alnajem@gmail.com; 2Department of Biochemistry, College of Medicine, Kuwait University, Safat 13110, Kuwait

**Keywords:** spermatogenesis, ischemia reperfusion injury, JAK2/STAT3, Keap1/Nrf2, apoptosis, animal model, male infertility

## Abstract

The male reproductive system requires the pleiotropic activity of JAK/STAT to maintain its function, especially spermatogenesis. The study aims to investigate the effect of JAK2 signaling on the expression of the Keap1/Nrf2 axis, spermatogenesis, and the Sertoli cells (Sc) junctions in an animal model of testicular ischemia reperfusion injury (tIRI). Testes subjected to tIRI exhibited increased JAK2/STAT3 activity associated with spermatogenic arrest and reduced expression of the Sc junctions. In addition, there was an increased protein expression of Keap1 and decreased Nrf2., which was coupled with the downregulation of gene expression of antioxidant enzymes. Reduced SOD and CAT activities were accompanied by increased lipid peroxidation and protein carbonylation during tIRI. Increased caspase 9 activity and *Bax*/*Bcl2* ratio indicated initiation of apoptosis. Inhibition of JAK2 activity by AG490 maintained the integrity of spermatogenesis and SC junctions, normalized the expression of the Keap1/Nrf2 axis and its downstream antioxidant enzymes, and prevented germ cell apoptosis. The results further emphasized the regulatory role of JAK2/STAT3 on spermatogenesis, Keap1/Nrf2 signaling, and maintenance of the testicular redox balance to combat testicular dysfunction and male infertility.

## 1. Introduction

Spermatogenesis is a dynamic process culminating in the production of mature sperms [1]. It occurs in the testis’s seminiferous tubules (STs) under tight hormonal and enzymatic regulations. The structure of the supratesticular artery renders the testicular STs under constant low oxygen tension [2]. Interestingly, this semi-hypoxic microenvironment is well tolerated by spermatogenesis, which protects sperms from naturally produced intracellular reactive oxygen species (ROS). However, the testis is also vulnerable to any restrictions to the blood flow due to constriction of the testicular artery. This will jeopardize testicular function, especially spermatogenesis, making testicular oxidative stress the key culprit in most male infertility conditions [3,4,5]. Therefore, understanding the molecular mechanisms regulating testicular redox homeostasis and spermatogenesis will unravel the underlying causes of male fertility anomalies and identify therapeutic targets.

Testicular torsion (TT) is a urologic emergency that results in ischemic testis due to an acute block of blood flow. Surgical intervention is needed for the detorsion (D) of the spermatic cord and artery to restore blood flow to the ischemic testis. Depending on the duration of TT, testicular dysfunction and spermatogenic arrest could lead to male subfertility and/or infertility problems [6]. TTD-induced animal models were created to mimic the pathophysiology of the condition [7]. At the molecular level, in vivo, animal models established that TTD is presented as a testicular ischemia reperfusion injury (tIRI) [8]. At the cellular level, disrupted ST structure and arrested spermatogenesis were clear phenotypes of experimental tIRI. Scientific investigations identified oxidative stress, inflammation, and germ cell apoptosis as the key underlying causes of these animal models’ observed testicular dysfunction [8]. Extensive studies reported on the effects of natural and synthetic drugs in attenuating the deleterious consequences of tIRI [9]. However, evidence for the signaling network that governs spermatogenesis and leads to its arrest during tIRI is limited.

The testis employs the Janus kinase/signal transducer and activator of transcription (JAK/STAT) as its central signaling regulatory pathway [10,11]. It protects the sensitive and complex nature of spermatogenesis and safeguards spermatozoa against ROS toxicity, which weakens their fertilization competence [12]. It also plays a major role in germ stem cell maintenance and maturation of functional sperms [12]. The JAK/STAT pathway propagates its signals via transcriptional activation of specific genes [11]. As a protective mechanism, the testis additionally maintains a tight transcriptionally-regulated antioxidant system to counteract physiologically generated ROS. The nuclear factor E2-related factor 2 (Nrf2) is a redox-sensitive transcription factor that has an important role in cellular adaptation to oxidative stress [13]. Nrf2 is the master gene transcription regulator of several antioxidant enzymes like catalase (CAT), superoxide dismutase (SOD), glutathione reductase (GSR), and quinone reductase 1 (NQO1). Physiologically, Nrf2 is bound to its negative regulator, Keap1 (Kelch-like ECH-associated protein 1). Keap1 sequesters Nrf2 in the cytoplasm and enhances its ubiquitination and degradation. However, under oxidative stress, Keap1 is inhibited by modifications at its cysteine residues, allowing the release of Nrf2. Upon its translocation to the nucleus, Nrf2 binds directly to antioxidant response elements (AREs) located in the promoter region of genes encoding antioxidants and multiple phase 2 detoxifying proteins/enzymes, causing their activation [14]. In the Keap1/Nrf2 axis, Keap1 is the electrophile/ROS sensor, while Nrf2 is the effector protein [15,16]. The Keap1/Nrf2 signaling pathway plays a vital role in maintaining reproductive function in mammals [17]. In the testis, Nrf2 expression was histologically detected in the cytoplasm and nucleus of spermatogonia and sperm cells [17]. *Nrf2*-knockout mice were presented with an oxidative disturbance in spermatogenesis emphasizing its role in the process of sperm production [18]. Furthermore, it was suggested that activation of the MAPK/Nrf2 pathway could damage the blood–testis barrier integrity and cause spermatogenic cell apoptosis [19]. Thus, Nrf2 expression could act as a predictive marker for testicular dysfunction and male infertility. Furthermore, its key antioxidant role and implications in several body system pathologies [20,21] and some cancer types [22,23] emphasize its promising use as a pivotal therapeutic target for future treatment modalities.

Several studies demonstrated a coordinated expression of JAK and Nrf2 in various disease conditions but without any evidence for direct regulation [24]. In this study, we sought to identify whether the JAK2/STAT3 signaling exerts direct regulation over the Keap1/Nrf2 pathway, spermatogenesis, and expression of the Sertoli cells (Sc) junctions.

## 2. Materials and Methods

### 2.1. Rat Model of tIRI and Drug Treatment

A 12 h light/12 h dark cycle was used for thirty-six male Sprague–Dawley rats (8 weeks old, 250–300 g) with food and water supplied ad libitum. Each rat was shaved at the ilioinguinal region, and betadine and 70% ethanol were used for disinfection. Thereafter, anesthesia (ketamine 50 mg/kg and xylazine 2 mg/kg) was injected intraperitoneally (i.p.). The animals were randomly assigned to 3 groups (*n* = 12 each); sham, unilateral tIRI, and tIRI treated with AG490, a JAK inhibitor. The sham rats underwent an ilioinguinal incision at the left side, exposing the left testis for 60 min followed by replacing it into the scrotal sac and suturing the incision. After 4 h, the sham rats were humanely sacrificed by decapitation. The tIRI rats had a unilateral ischemic injury to the left testis for 60 min using a straight bulldog clamp, applying 700 g pressure to completely occlude the blood flow from the testicular artery to the testis. The rats were i.p. injected with 250 µL dimethyl sulfoxide (DMSO, drug vehicle) 30 min before testicular reperfusion for 4 h followed by animal sacrifice. The AG490-treated rats underwent the same procedure as the tIRI group, except this group was injected with AG490 (40 mg/kg dissolved in DMSO; Sigma-Aldrich, St. Louis, MO, USA) instead of DMSO [25]. All three groups’ contralateral (right) testes were used as a positive internal control. All dissected testes were immediately preserved according to the nature of the downstream experiments.

### 2.2. Hematoxylin and Eosin Staining

Harvested testes were fixed in Bouin’s fixative and embedded in paraffin blocks. The 4 µm testicular tissue sections were stained with hematoxylin and eosin (H&E) for spermatogenesis assessment using the Johnson scoring method under light microscopy [25,26]. Images were taken at 10×, 20×, and 40× magnifications. The absence of germ cells in the seminiferous tubules was used to score spermatogenesis in the following order: spermatozoa (10–8) > spermatid (7–6) > spermatocytes (5–4) > spermatogonia (3) > Sertoli cells (2) > no cells.

### 2.3. Western Blot Analysis

Total protein extracts were prepared from the harvested testes using radio-immunoprecipitation assay (RIPA) buffer (Sigma-Aldrich, St. Louis, MO, USA) and stored at −80 °C. The primary antibodies for JAK2 (ab108596) and ph-JAK2 (ab195055) were purchased from abcam (Cambridge, UK). The primary antibodies for STAT3 (mAb#4904) and ph-STAT3 (mAb#9145) were purchased from Cell Signaling Technology (Danvers, MA, USA). The primary antibodies for Nrf2 (MBS9600480) and Keap1 (MBS2536215) were purchased from MyBioSource.com (San Diego, CA, USA). Protein extracts (150 μg) were resolved through 10% SDS-PAGE. Separated proteins were transferred to a PVDF membrane followed by blocking and incubation with primary antibody for each target protein individually. The primary antibody dilutions are as follows: JAK2 (1:1000), ph-JAK2 (1:500), STAT3 (1:1000), ph-STAT3 (1:500), Nrf2 (1:1000), and Keap1 (1:1000). The PVDF membrane was then treated with a horseradish peroxidase/HRP-conjugated secondary antibody. Protein band signal amplification was performed using the electrochemiluminescence method (ECL) (Thermo Fisher Scientific, Waltham, MA, USA) and visualized by the ChemiDoc™ MP Imaging System (BioRad, Hercules, CA, USA). Band intensity quantification was measured using the Image Lab software v. 4.1 (BioRad, Hercules, CA, USA).

### 2.4. Enzyme-Linked Immunosorbent Assay

Enzyme-linked immune-sorbent assay (ELISA) was utilized to quantitate the protein expression of the Sc junctions: connexin-43 (Cx-43; ER0881), occludin (Ocln; ER1206), claudin-11 (Cldn11; E11717r), zonula occludens-1/Tight junction protein 1 (ZO-1/TJP1; ER1386) were purchased from FineTest (Wuhan, Hubei, China). Experimental and standard curve samples were loaded onto the washed 96-well ELISA plate. Proteins were detected using their respective biotin-labeled antibodies. Thereafter, the 3,3′,5,5′-Tetramethylbenzidine (TMB) substrate solution was added for signal amplification, and signal OD absorbance was immediately read at 450 nm. The proteins’ concentrations were interpolated from the standard curve.

### 2.5. Biochemical Assays

The SOD and CAT colorimetric activity assay kits (Invitrogen, Waltham, MA, USA) were utilized. The activities of the two antioxidant enzymes were measured in 150 μg of total protein extracts using a microplate reader. A generated standard curve was used to calculate the antioxidant enzyme activities in the samples and controls, which were corrected for the dilution factor.

Malondialdehyde (MDA) is a lipid peroxidation product assayed by the BIOXYTECH^®^ LPO-586™ kit (Oxis Research, Portland, OR, USA). The MDA assay was performed following the manufacturer’s protocol. Protein samples with standardized concentrations were loaded and processed using 96-well plates. The MDA sample absorbance was read at 586nm. Finally, the MDA concentration was calculated using a specific formula provided by the manufacturer’s protocol.

The NADP/NADPH Quantification Kit (Sigma-Aldrich, St. Louis, MO, USA) was used as an indirect measure for NADPH oxidase activity and the rate of ROS generation. Following the manufacturer’s protocol, the concentrations of total NADP and NADPH were read at 450 nm and calculated using a standard curve. The NADP/NADPH ratio was calculated as (NADPt—NADPH)/NADPH.

### 2.6. Relative mRNA Expression by Real-Time PCR

RNA extraction from frozen testicular tissue samples was performed by homogenization in Trizol following the manufacturer’s guidelines (Invitrogen, Carlsbad, CA, USA). Total RNA was reverse transcribed to complementary DNA (cDNA) using the high-capacity cDNA reverse transcription kit (Life Technologies, Grand Island, NY, USA). For real-time PCR, the cDNA template was loaded on a 96-well plate before adding the 2× PCR master mix and 20× gene-specific Taqman assay (Thermo Fisher Scientific, Waltham, MA, USA). The mRNA expression was measured for the following genes: *Sod* (Rn00566938_m1), *Cat* (Rn00560930_m1), *Nqo-1* (Rn00566528_m1), *Gsr* (Rn01482159_m1), *Bax* (Rn01480161_g1) and *Bcl2* (Rn99999125_m1). The Taqman assay for β-actin was used as an endogenous control. Amplification reactions were performed using the Quant Studio System™ 5 Real-Time PCR Instrument (Applied Biosystems, Waltham, MA, USA) using the machine’s standard protocol. The relative mRNA expression for individual genes was calculated using the 2^−ΔΔCT^ method [27].

### 2.7. Statistical Analysis

GraphPad Prism v8.0 (GraphPad Software Inc., San Diego, CA, USA) was used to analyze the raw data. Experimental group data were subjected to a one-way analysis of variance (ANOVA) followed by the Holm–Sidak multiple comparisons test. The data were represented as mean ± standard deviation (SD) and the *p*-value for significance was set at <0.05.

## 3. Results

### 3.1. Activation of JAK2/STAT3 during tIRI

The phosphorylation of the JAK2 and STAT3 was enhanced by 1.87- and 3.85-fold, respectively, in the ipsilateral testes of tIRI-subjected testes compared to sham testes suggesting pathway activation (Figure 1). Treatment with AG490 normalized the phosphorylation levels of JAK2 and STAT3. There was no significant change in the phosphorylation ratio of the JAK2/STAT3 in the contralateral testes of the three rat groups (*p*-value = 0.0138).

### 3.2. JAK2/STAT3 Regulates Spermatogenesis

Histologically, the tIRI-subjected ipsilateral testes elicited negative histopathological alterations of spermatogenesis with perturbed stages (Figure 2A–C). Conversely, the seminiferous epithelium showed normal germ cell distribution in both sham and AG490-treated groups (Figure 2D,E, respectively). An average Johnsen score of 6.83 ± 0.75 indicated the absence of spermatozoa and the presence of spermatids only suggesting arrested spermatogenesis without spermiation (Figure 2F). AG490 treatment prevented spermatogenic arrest implying the role of JAK2/STAT3 in sperm maturation as observed at the morphological level. All contralateral testes displayed normal histological stages of spermatogenesis (*p*-value > 0.05).

### 3.3. JAK2/STAT3 Regulates the Protein Expression of Sc Junctions

The protein expression of Cldn11, Cx43, and ZO1 was reduced by 1.9-, 1.5- and 1.6-fold, respectively, in tIRI-subjected testes compared to sham (Figure 3). In contrast, the protein expression of Ocln was increased by 1.5 fold in tIRI compared to the sham. The expression of the four Sc junction proteins was normalized with AG490 treatment. There was no significant change measured in all the contralateral testes (*p*-value > 0.05).

### 3.4. JAK2/STAT3 Regulates the Keap1/Nrf2 Axis and its Downstream Gene Targets

In comparison to sham testes, the tIRI-subjected testes showed a 2.86-fold increase in the protein expression of Keap1 in contrast to the 1.82-fold decrease in Nrf2 protein expression (Figure 4). AG490 treatment normalized the protein expression of Keap1 and Nrf2. No significant difference was observed between the contralateral testes for Keap1 (*p*-value > 0.05).

In addition, the mRNA expression for antioxidant enzymes regulated by Nrf2 was measured (Figure 5). In tIRI-subjected testes, the mRNA expression of *Sod*, *Cat*, *Nqo1*, and *Gsr* was decreased by 7.7-, 7.1-, 2.6-, and 3.8-fold, respectively, when compared to the sham testes. This was normalized by AG490 treatment. There was no statistical significance between the three contralateral testes for all the aforementioned antioxidant genes.

### 3.5. JAK2/STAT3 Regulates Oxidative Stress

The enzyme activities of SOD and CAT were decreased by 1.2- and 1.6-fold in the tIRI-subjected testes compared to the sham testes (Figure 6). The concentration of the lipid peroxidation marker MDA was increased by 1.36-fold compared to sham testes. There was a 2.85-fold increase in the NADP to NADPH ratio in the tIRI-subjected testes compared to the sham testes. This is an indirect indicator of increased NADPH oxidase (NOX) activity and hence the rate of ROS generation. The AG490-treated rats exhibited normal values for the above oxidative stress parameters that are comparable to sham levels. The contralateral testes from the three experimental groups showed no statistical significance among them (*p*-value > 0.05).

### 3.6. JAK2/STAT3 Regulates Apoptosis Switches

During tIRI, the mRNA expression of the anti-apoptosis gene *Bcl2* was downregulated by 6.6-fold, while the pro-apoptosis *Bax* mRNA expression was upregulated by 5-fold in the ipsilateral testes in comparison to the sham group (Figure 7). A significant increase in the *Bax*/*Bcl2* ratio by 44-fold was calculated confirming induction of apoptosis in the tIRI-subjected testes compared to the normal ratio in the sham group. The AG490 treatment normalized the modulated mRNA expression of *Bax* and *Bcl2* and hence their ratio. There was no significant change in the transcription levels of *Bax* and *Bcl2* genes in the contralateral testes from the three experimental groups (*p*-value > 0.05).

## 4. Discussion

The health of the male reproductive system depends on the activity of the JAK/STAT signaling pathway. In the testis, it regulates germline sexual development and stem cell self-renewal ensuring a continuous sperm supply and sperm function such as capacitation and motility. JAK2 is a redox-sensitive kinase that is activated during oxidative stress and a known pathogenic factor in many diseases including those of the male reproductive system. The outcome of this study provides evidence for JAK’s regulation of spermatogenesis, protein expression of Sc junctions, and the Keap1/Nrf2 axis during tIRI-induced oxidative stress.

The JAK/STAT signaling pathway is central for the testis to fulfill its proliferative demand for continuous sperm production. During the early developmental stages of the testis, the JAK/STAT pathway is the mediator of key signals produced by the somatic gonad cells that regulate the development of male germ cells [28]. It was reported that the JAK2 gene expression is respectively altered to correspond to the status of the testis, which links its activity to the homeostasis of the testis [29]. Furthermore, the survival of germ cells in the presence of reproductive toxins was attributed to the activation of the JAK/STAT pathway [30]. Expression of members of the JAK/STAT proteins in the active sperm and their role in fertilization suggest their relevance to male factor infertility if disrupted [31]. While JAK1 is localized to the equatorial region and midpiece of the sperm, JAK2 is found in the sperm tail. During capacitation, JAK1/2 were activated despite their differential localization in the sperm [32]. Bioinformatics and in vivo studies confirmed a correlation between the expression of proteins in the JAK pathway with hypoxia and oligozoospermia [33]. JAK was recently identified in a whole-genome profile as a new variant associated with teratozoospermia, indicating its involvement in male infertility [34]. However, data from bioinformatic analyses need in vivo research to investigate the exact role of JAK isoforms in spermatogenesis and sperm function. In this study, an average Johnsen score of 6.8/10 indicates the arrest of spermatogenesis at the spermatids stage and complete loss of spermatozoa during tIRI. This confirms that early stages of spermiogenesis are sensitive to acute moderate reduction in blood flow and could have a large impact on sperm maturation [35]. Inhibition of JAK2 phosphorylation and subsequent inactivation prevented damage to the ST structure and protected spermatogenesis during tIRI. Thus, JAK2 activation during testicular oxidative stress and triggering apoptosis is considered a protective mechanism to preserve overall testicular function. This further confirms that JAK2 expression is modulated in parallel to the changes in the testis microenvironment in support of maintaining balanced testis homeostasis during adverse pathological conditions such as tIRI [29].

During spermatogenesis, differentiated spermatogonial stem cells are transported from the basal to the apical compartment to undergo meiosis and eventually form mature spermatozoa. The movement of developing germ cells through the ST is governed by specialized cell junctions [36]. The seminiferous epithelium contains three main tight junction (TJ) proteins: Ocln, Cldn11, and ZO-1. Impaired spermatogenesis is tightly associated with defects in these TJ proteins, confirming their combined effect in male-factor subfertility or infertility. Analysis of gene knockout mouse models for these TJ proteins revealed their vital role in spermatogenesis and male fertility [37]. Ocln was initially recognized as a key player in the TJ assembly. However, the presence of polarized epithelial cells with TJs in Ocln-deficient embryonic stem cells suggested other roles for Ocln [38]. This was confirmed by the presence of a normal BTB in ocln-null mice [39]. In addition, the testis of homozygous *Ocln*-null mice generated from genetically modified spermatogonial stem cells showed impaired spermatogenesis and decreased fertility [40]. Interestingly, injection of an Ocln peptide directly into the testis induced reversible aspermatogenesis [41]. This suggests that Ocln overexpression may weaken its interaction with adherens junctions leading to the premature release of germ cells from the ST epithelium. Lastly, Ocln is thought to have a regulatory role based on its interactions with other ST kinases as opposed to a mere structural component of the TJ fibrils [42]. This is in line with the current observation that tIRI-induced Ocln expression is associated with the activation of the JAK2/STAT3 signaling pathway and spermatogenic arrest. Cldn11 is overexpressed during stages V-VII, which precedes the movement of preleptotene/leptotene spermatocytes across the BTB [43]. The testis of Cldn11-null mice was presented with a weak paracellular barrier leading to the complete absence of spermatozoa [44]. A later study showed that *Cldn11*-null mice were infertile due to the lack of spermatozoa since germ cells were unable to differentiate beyond the spermatocyte stage [45]. Both Ocln and Cldn11 require the adaptor protein ZO-1 to attach to the actin cytoskeleton during TJ strand assembly. ZO-1 is suggested to promote claudins’ polymerization to TJs; thus, its partial loss could disable polarized epithelial cells from assembling TJs [46]. Interestingly, while the knockout mice for Cldn11 and Ocln are viable and infertile, the *Tjp1*-null mice are embryonically lethal [37]. ZO-1 is localized to the apical ectoplasmic specialization flanking early elongating spermatids before sperm release, where it interacts with Cx43, the prominent gap junction protein [47]. Therefore, it was suggested that disruption of ZO-1/Cx43 interaction could affect spermatid movement during spermiation at stages IV-VI of spermatogenesis [37]. Although the BTB structure was not affected, the Cx43-null mice were infertile with arrested spermatogenesis and diffused Cldn11 expression [48]. In this context, the absence of mature spermatozoa induced by tIRI could be attributed to the downregulated expression of Cldn11, ZO-1, and Cx43 due to their prominent localization at the apical ectoplasmic specialization. This effect was abolished by inhibiting JAK2 activation suggesting its regulation of early stages of spermiogenesis.

A key risk factor to the health of spermatogenesis is oxidative stress. Spermatogenesis produces ROS as a side product of high mitochondrial oxygen consumption. However, these ROS are physiologically eliminated by an extensive antioxidant system in testicular cells. Yet, the testis is still vulnerable to any excessive ROS generation triggered by internal or external factors [49]. Oxidative stress has multiple consequences on testicular viability as it causes sperm DNA damage, membrane lipid peroxidation, decreased motility, and spermatogenesis dysfunction [50]. The strong impact of oxidative stress on male infertility led to the invention of the term “Male oxidative stress infertility (MOSI)” [51]. The induction of spermatogenic arrest is mediated through a cascade of cellular signaling pathways. We previously demonstrated that the JAK2/STAT3 signaling pathway is directly involved in ROS-induced DNA damage and modulation of the DNA repair pathways in germ cells [25]. Here we further demonstrate that inhibition of tIRI-activated JAK2 decreases ROS production, prevents lipid oxidation, restores SOD and CAT activities, and identifies Keap1/Nrf2 as its downstream target axis in its signaling mechanism.

The transcription factor Nrf2 is the master regulator for basal and inducible transcription of cellular antioxidant enzymes. Nrf2-null mice showed a progressive decline in the expression of antioxidant enzymes, diminished sperm count and motility, elevated levels of lipid peroxidation, and germ cell apoptosis [18]. In addition, single nucleotide polymorphism in the *Nrf2* gene was associated with the severity of sperm DNA damage and impaired sperm motility and count in infertile men [52]. This further emphasizes the functional role of Nrf2 in the antioxidant mechanism of spermatogenesis. In a Chinese population, low seminal parameters were associated with low *Nrf2* mRNA expression in subfertile men [53]. An in vivo model of testicular injury demonstrated that spermatogenesis dysfunction was rescued by overexpression of Nrf2 and Cx43 [54].

Exploring the Nrf2-ome website revealed a high functional overlap between JAK and Nrf2 giving support to some experimental data [55]. The website predicted that STAT1 is a likely transcription factor of Nrf2, while STAT3 is a possible Nrf2 interactor. Using a myocardial IRI model, inhibition of STAT3 by AG490 blocked the nuclear translocation of Nrf2 and subsequent activation of the genes encoding HO-1 and SOD enzymes [56]. Their claim was based on the possible connection between Nrf2 and the JAK/STAT pathway as proposed by the Nrf2-ome website. Similarly, other studies suggested that Nrf2 upregulation blocked the signaling of the JAK/STAT pathway via STAT3 inhibition [57,58]. Data from recent studies only showed the presence of an inverse relationship between the activities of JAK and Nrf2 during oxidative stress. In testis subjected to ionizing radiation, damage to spermatogenesis was preserved via Nrf2 activation leading to reduced p-JAK1/p-STAT3 levels [59]. A previous study from our group also suggested the presence of an inverse expression of JAK2 and Nrf2 during tIRI [60]. Furthermore, heavy metal-induced oxidative stress was found to inhibit JAK/STAT activation and enhance Nrf2 expression [61]. Thus, the signaling mechanism governing the interplay between Nrf2 and JAK/STAT pathways during oxidative stress could be tissue-dependent, but this remains controversial [62].

The constitutive expression of the transcription factor Nrf2 in the testis highlights its role in oxidative stress and metabolic homeostasis [17,63,64]. In health and disease, Nrf2 regulates the gene transcription of phase 2 antioxidant and redox cycling enzymes by binding to the ARE DNA element in their promoter region [65,66,67]. Several studies reported on the effect of antioxidant treatments on preserving testicular function by upregulating the expression of Nrf2 and its subsequent target genes like *Gsh, Nqo1, Sod,* and *Ho-1* [68]. Inhibitors of Nrf2, on the other hand, showed reduced expression of Nrf2 and its downstream antioxidant gene targets in the male reproductive system but did not necessarily affect their respective enzyme activity [68]. It was earlier demonstrated that SOD activity in rat testis decreases as spermatogenesis progresses with the lowest activity measured in spermatozoa [69]. Similarly, low Cat mRNA expression and hence less enzyme activity were present in meiotic and post-meiotic cells, but more abundant in somatic testicular cells [70]. Collectively, the natural low abundance and activity of SOD and CAT in spermatozoa render them extremely vulnerable to even a slight decrease in blood flow and ROS-induced oxidative stress. Nrf2-null mice had significantly reduced mRNA expression and activity of antioxidant enzymes [18]. In addition, there was a substantial increase in the concentration of reactive lipid peroxidation products like MDA, HAE, and 4-HNE. This is partly attributed to the rich PUFA content in the sperm membrane, which increases sperm’s vulnerability to the least increase in ROS levels. In some pathologies associated with decreased Nrf2 expression, many of its target genes are responsible for preventing lipid peroxidation; however, they are electrophilically modified by lipid peroxidation products [71]. It was earlier reported that several reactive lipid products form adducts in Keap1, modifying the key cysteine residues (C273 and C288), and causing its inactivation and release of Nrf2 [72]. Such an effect will significantly enhance lipid peroxidation and disease progression. These reports are in agreement with the current findings of increased ROS generation and MDA levels, accompanied by transcriptional downregulation of the antioxidant genes and decreased antioxidant enzyme activities. Such effects were reversed by JAK2 inhibition. This suggests that tIRI-induced oxidative stress, albeit transient, the testicular redox system is disrupted rendering germ cells oxidatively defenseless and vulnerable to ROS.

Induction of apoptosis is one of the testis responses to acute oxidative stress injuries. This protective mechanism preserves the cells’ energy and their surroundings from progressive oxidative damage. The activation of the JAK/STAT pathway in oxidative stress-mediated apoptotic cell death is well-recognized in testicular dysfunction including tIRI [10,25,73]. Similarly, Nrf2 stimulation in different IRI tissue models proves its ability to ameliorate IRI-induced oxidative stress [74]. Nrf2 was also shown to exert cytoprotective effects via its transcriptional upregulation of the antiapoptotic *Bcl-2* gene, downregulation of the pro-apoptosis proteins Bax and Bcl-XL, and a reduction in caspase 3/7 activity [75,76]. The identification of Nrf2 as a downstream target for JAK2/STAT3 signaling and modulating the apoptosis switches in the current tIRI animal model further confirms its critical role in promoting germ cell viability and testicular function.

## 5. Conclusions

JAK/STAT can be considered a quality control signaling pathway for cellular events in the male reproductive system. Its manipulation to protect spermatogenesis could provide novel therapies for male infertility in the clinic. There is a shortage of studies focusing on the JAK signaling mechanisms and its cross-talk with other cellular pathways in the testis. This study provides a novel regulatory mechanism for JAK2/STAT3 signaling via modulation of the Keap1/Nrf2 axis in preventing testicular oxidative stress. Its impact on different aspects of spermatogenesis necessitates careful exploration of molecular mechanisms to ensure spermatogenesis’s integrity and the ST’s structural integrity. Thus, the JAK2/STAT3 and Keap1/Nrf2 pathways and their components can be considered therapeutic targets for drug design to manage male reproductive system diseases.

## Figures and Tables

**Figure 1 cells-12-02292-f001:**
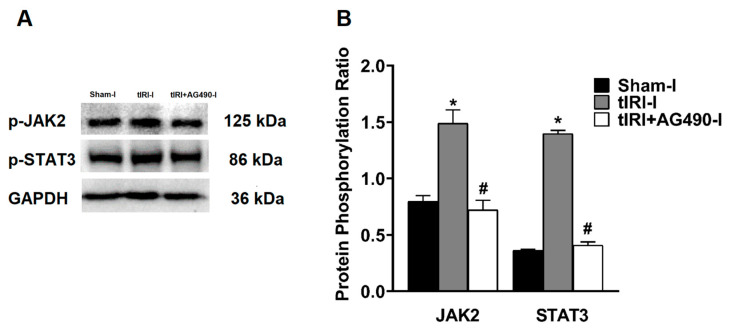
The effect of AG490 on the phosphorylation of JAK2 and STAT3. (**A**) Representative pictures of Western blots demonstrating the protein levels of the phosphorylated forms of JAK2 and STAT3. (**B**) Diagrammatic blot demonstrating the signal intensity quantification of the phosphorylation ratios of p-JAK2/JAK2 and p-STAT3/STAT3. The values are presented as the mean ± SD (*n* = 6/group), *p*-value < 0.05. * tIRI compared to sham and ^#^ AG490-treated compared to tIRI. Abbreviations: JAK = Janus kinase; STAT = signal transducer and activator of transcription; tIRI = testicular ischemia reperfusion injury; I = Ipsilateral.

**Figure 2 cells-12-02292-f002:**
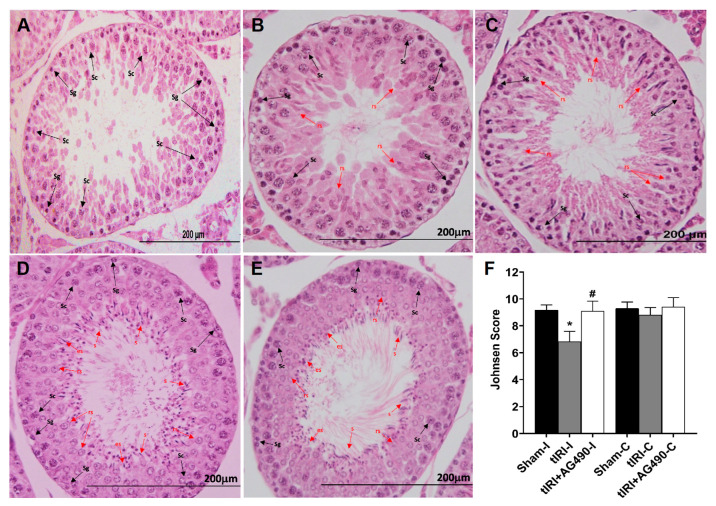
The effect of AG490 on spermatogenesis. The plate is a representation of H&E-stained testicular tissues from ipsilateral testes (**A**) A tIRI-subjected testis with a Johnsen score of 5 due to the absence of spermatozoa and spermatids and the presence of only spermatocytes and spermatogonium. (**B**) A tIRI-subjected testis with a Johnsen score of 6 as it lacks spermatozoa and elongated spermatids, but rather shows only a few round spermatids. (**C**) A tIRI-subjected testis with a Johnsen score of 7 due to the presence of many round spermatids but no elongated spermatids or spermatozoa. (**D**,**E**) Sections from sham and AG490-treated testes with an average Johnsen score of 9.1 showed normal spermatogenesis and germ cell distribution. (**F**) A diagrammatic plot demonstrating the average Johnsen scores for each experimental group. * tIRI compared to sham and ^#^ AG490-treated compared to tIRI. Abbreviations: C = contralateral, s = spermatozoa; es = elongated spermatids; rs = round spermatids; Sc = spermatocytes; Sg = spermatogonia. Images of histological sections were captured at 40× magnifications (scale bar 200 mm).

**Figure 3 cells-12-02292-f003:**
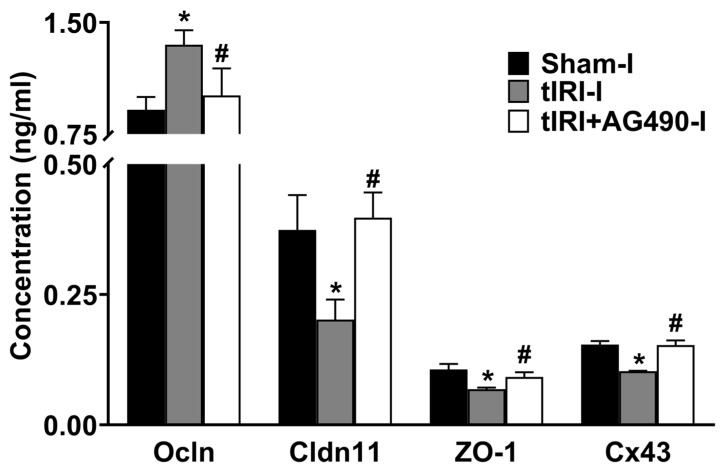
The effect of AG490 on the protein expression of Sc junctions. The protein concentration of Sc junctions in testicular tissue was measured using their respective ELISA assays. Data are presented as the mean ± SD (*n* = 6/group), *p*-value < 0.05. ***** tIRI compared to sham and **^#^** AG490-treated compared to tIRI. Abbreviations: Ocln = Occludin; Cldn11 = Claudin 11; ZO-1 = Zonula Occludens-1; Cx43 = Connexin 43.

**Figure 4 cells-12-02292-f004:**
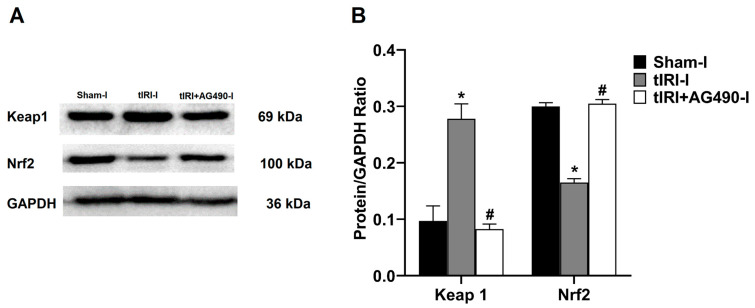
The effect of AG490 on the protein expression of Keap1 and Nrf2. (**A**) Representative pictures of Western blots demonstrating the protein expression of Keap1 and Nrf2. (**B**) Quantification of Keap1 and Nrf2 signal intensity plotted as their ratio to GAPDH. The values are presented as the mean ± SD (*n* = 6/group), *p*-value < 0.05. * tIRI compared to sham and ^#^ AG490-treated compared to tIRI. I = Ipsilateral. Abbreviations: Keap1 = Kelch-like ECH-associated protein 1; Nrf2 = nuclear factor E2-related factor 2; GAPDH = glyceraldehyde 3-phosphate dehydrogenase.

**Figure 5 cells-12-02292-f005:**
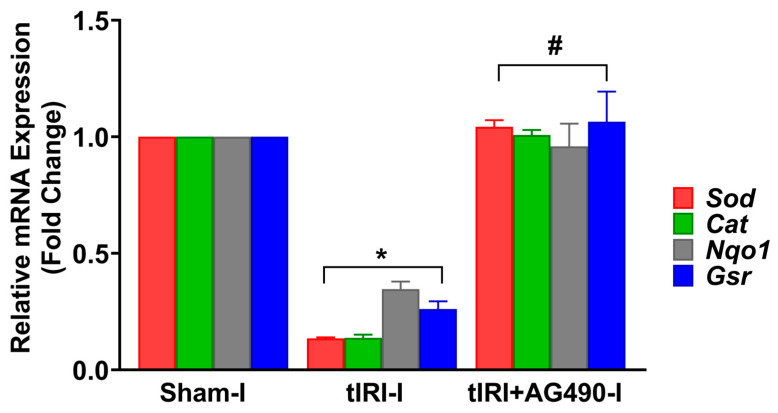
The effect of AG490 on the gene expression of Nrf2 downstream target genes. The relative mRNA expression of *Sod*, *Cat*, *Nqo1*, and *Gsr* was measured by the two-step reverse transcription and real-time PCR. The fold change in gene expression for tIRI and tIRI+AG490 groups was calculated in relation to the sham group using the 2^−ΔΔCt^ method. Data are presented as the mean ± SD (*n* = 6/group), *p*-value < 0.05. * tIRI compared to sham and ^#^ AG490-treated compared to tIRI. Abbreviations: *Sod* = superoxide dismutase gene; *Cat* = catalase gene; *Nqo1* = quinone reductase gene; and *Gsr* = glutathione reductase.

**Figure 6 cells-12-02292-f006:**
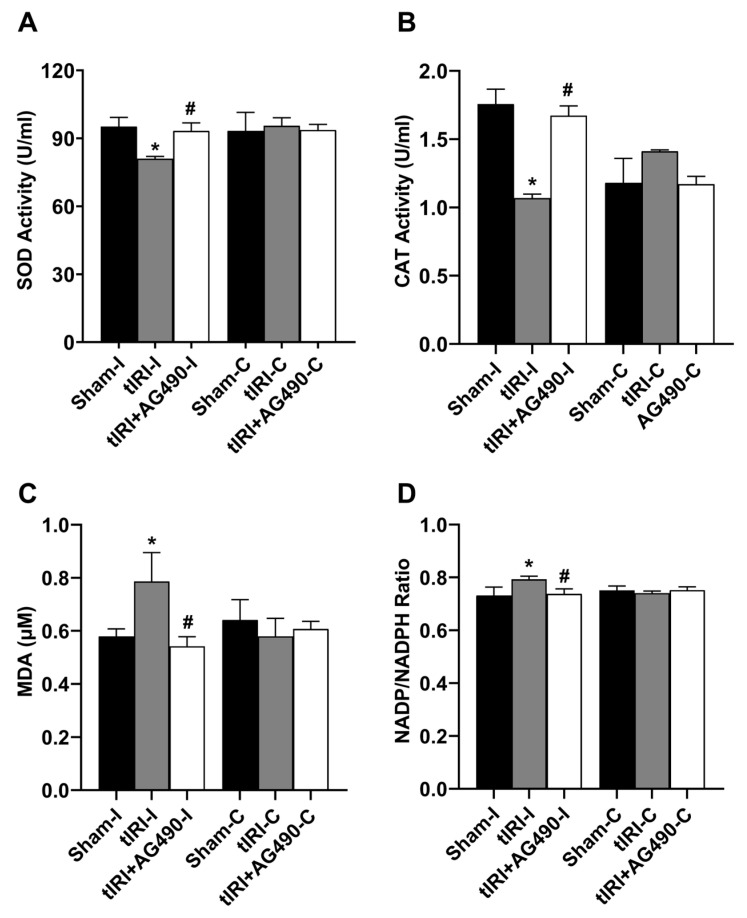
The effect of AG490 on oxidative stress parameters. The enzyme activity of (**A**) SOD and (**B**) CAT and the levels of (**C**) MDA and (**D**) NADP/NADPH ratio were measured using their respective biochemical assays. Data are presented as mean ± SD (*n* = 6/group), *p*-value < 0.05. * tIRI compared to sham and ^#^ AG490-treated compared to tIRI. SOD = superoxide dismutase; CAT = catalase; MDA = malondialdehyde; NADP = nicotinamide adenine dinucleotide phosphate; NADPH = reduced NADP.

**Figure 7 cells-12-02292-f007:**
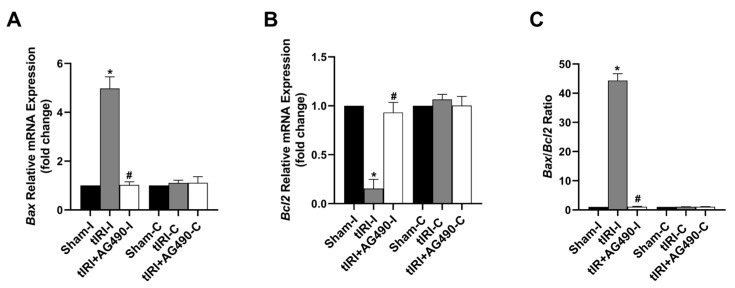
The effect of AG490 on the gene expression of the apoptosis regulators Bax and Bcl2. The relative mRNA expression of (**A**) *Bax* and (**B**) *Bcl2* was measured by the two-step reverse transcription and real-time PCR. The fold change in gene expression for tIRI and tIRI+AG490 groups was calculated in relation to the sham group using the 2^−ΔΔCt^ method. (**C**) The changes in the *Bax*/*Bcl2* ratio. Data are presented as the mean ± SD (*n* = 6/group), *p*-value < 0.05. * tIRI compared to sham and ^#^ AG490-treated compared to tIRI. Abbreviations: *Bcl2* = B-cell lymphoma protein 2 gene; *Bax* = Bcl2-associated protein × gene.

## Data Availability

The raw data supporting the conclusions of this article will be made available by the authors, without undue reservation.

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
