# Peer review of "The Regulatory Effects of JAK2/STAT3 on Spermatogenesis and the Redox Keap1/Nrf2 Axis in an Animal Model of Testicular Ischemia Reperfusion Injury"

_cells, 2023, doi:10.3390/cells12182292_

Round 1
Reviewer 1 Report
In this study authors investigated the effect of JAK2 signaling on the expression of the Keap1/Nrf2 axis, spermatogenesis, and the Sertoli cells junctions in an animal model of testicular ischemia reperfusion injury.
The study is very interesting and generally well written but it presents some flaws that must be resolved.
Lines 62-63: the role of NRF2 has been too semplified. In fact, it deserves to be pointed out that NRF2/KEAP1 signaling pathway plays a key role in several pathology (see these recent reviews PMID: 35453348, 36289931, 37519172, 36632321). This is an interesting point to add since the interesting results obtained by the authors may suggest further studies also in these pathologies.
2.3. Western Blot Analysis: Antibodies product codes and dilutions must be reported
Lines 111-112: did the authors load a gel with 150 mg of total protein ?
2.4. Enzyme-linked Immunosorbent Assay: ELISA kit product codes must be reported
Figure 1: did the authors evaluate total STAT3 and total JAK2? In figure A, authors must add the labeling on the lanes in order to understand what has been loaded in each lane
Figure 4A: authors must add the labeling on the lanes in order to understand what has been loaded in each lane
Acronyms must be written in full length when mentioned for the first time
Author Response
Response to Reviewer 1 Comments
In this study authors investigated the effect of JAK2 signaling on the expression of the Keap1/Nrf2 axis, spermatogenesis, and the Sertoli cells junctions in an animal model of testicular ischemia reperfusion injury.
The study is very interesting and generally well written but it presents some flaws that must be resolved.
The authors are grateful to the editor and reviewer for their precious time spent reviewing the manuscript and providing valuable comments. The authors have carefully considered the comments and addressed each one of them. We hope the revised manuscript meets your high standards.
Lines 62-63: the role of NRF2 has been too semplified. In fact, it deserves to be pointed out that NRF2/KEAP1 signaling pathway plays a key role in several pathology (see these recent reviews PMID: 35453348, 36289931, 37519172, 36632321). This is an interesting point to add since the interesting results obtained by the authors may suggest further studies also in these pathologies.
Authors Response
Based on the reviewer’s suggestion, the following sentences were added at line 80:
Thus, Nrf2 expression could act as a predictive marker for testicular dysfunction and male infertility. Furthermore, its key antioxidant role and implications in several body system pathologies [20,21] and some cancer types [22,23] emphasize its promising use as a pivotal therapeutic target for future treatment modalities.
- Abdul-Muneer, P.M. Nrf2 as a Potential Therapeutic Target for Traumatic Brain Injury. J Integr Neurosci 2023, 22, 81.
- Bukke, V.N.; Moola, A.; Serviddio, G.; Vendemiale, G.; Bellanti, F. Nuclear factor erythroid 2-related factor 2-mediated signaling and metabolic associated fatty liver disease. World J Gastroenterol 2022, 28, 6909-6921.
- Tossetta, G.; Fantone, S.; Montanari, E.; Marzioni, D., Goteri, G. Role of NRF2 in Ovarian Cancer. Antioxidants 2022, 11, 663.
- Ghareghomi, S.; Habibi-Rezaei, M.; Arese, M.; Saso, L.; Moosavi-Movahedi, A.A. Nrf2 Modulation in Breast Cancer. Biomedicines 2022, 10, 2668.
2.3. Western Blot Analysis: Antibodies product codes and dilutions must be reported
Authors Response
Section 2.3. Western blot was edited (below) to show the catalog numbers of primary antibodies and their dilutions.
2.3. Western Blot Analysis
Total protein extracts were prepared from the harvested testes using radio-immunoprecipitation assay (RIPA) buffer (Sigma-Aldrich, St. Louis, Mi, USA) and stored at -80oC. The primary antibodies for JAK2 (ab108596) and ph-JAK2 (ab195055) were purchased from abcam (Cambridge, UK). The primary antibodies for STAT3 (mAb#4904) and ph-STAT3 (mAb#9145) were purchased from Cell Signaling Technology (Danvers, MA, USA). The primary antibodies for Nrf2 (MBS9600480) and Keap1 (MBS2536215) were purchased from MyBioSource.com (San Diego, CA, USA). Protein extracts (150 mg) were resolved through 10% SDS-PAGE. Separated proteins were transferred to a PVDF membrane followed by blocking and incubation with primary antibody for each target protein individually. The primary antibody dilutions are as follows: JAK2 (1:1000), ph-JAK2 (1:500), STAT3 (1:1000), ph-STAT3 (1:500), Nrf2 (1:1000), and Keap1 (1:1000). The PVDF membrane was then treated with a horseradish peroxidase/HRP-conjugated secondary antibody. Protein band signal amplification was done using the electrochemiluminescence method (ECL) (Thermo Fisher Scientific, Waltham, MA, USA) and visualized by the ChemiDoc™ MP Imaging System (BioRad, Hercules, CA, USA). Band intensity quantification was measured using the Image Lab software (BioRad, Hercules, CA, USA).
Lines 111-112: did the authors load a gel with 150 mg of total protein ?
Authors Response
This was a typographical error and was corrected to “Protein extracts (150 mg) were resolved through 10% SDS-PAGE.”
2.4. Enzyme-linked Immunosorbent Assay: ELISA kit product codes must be reported
Authors Response
Section 2.4. Enzyme-linked Immunosorbent Assay was edited (below) to show the catalog numbers of ELISA kits and their manufacturer.
connexin-43 (Cx-43; ER0881), occludin (Ocln; ER1206), claudin-11 (Cldn11; E11717r) and zonula occludens-1/Tight junction protein 1 (ZO-1/TJP1; ER1386) were purchased from FineTest (Wuhan, Hubei, China).
Figure 1: did the authors evaluate total STAT3 and total JAK2? In figure A, authors must add the labeling on the lanes in order to understand what has been loaded in each lane
Authors Response
The authors evaluated total and phosphorylated forms of STAT3 and JAK2. The animal group labels were added to Figure 1A.
Figure 4A: authors must add the labeling on the lanes in order to understand what has been loaded in each lane
Authors Response
The animal group labels were added to Figure 4A.
Acronyms must be written in full length when mentioned for the first time
Authors Response
The manuscript was revised and edited to ensure that all acronyms were written in full length when mentioned for the first time.

Reviewer 2 Report
The focus of the manuscript is clear and it is well written. The data presented add some info to regulation of spermatogenesis.
My only concern regards fig.5 in which the dara of sham group have no standard deviation.
Author Response
Response to Reviewer 2 Comments
The focus of the manuscript is clear and it is well written. The data presented add some info to regulation of spermatogenesis.
The authors are grateful to the editor and reviewer for their precious time spent reviewing the manuscript and providing valuable comments. The authors have carefully considered the comments and addressed each one of them. We hope the revised manuscript meets your high standards.
My only concern regards fig.5 in which the dara of sham group have no standard deviation.
Authors Response
The calculation method for relative quantitative real-time PCR data sets was reported by Livak and Schmittgen [27]. In this method, it is commonly practiced to designate one sample, called the “calibrator sample” or “reference sample,” to calibrate the data set. The current study uses the “Sham” group as the calibrator sample. The calibrator sample value is normalized to itself, resulting in an RQ of 1, and to all other sample values in the experiment, producing RQ values relative to 1 without any error. Calibrating to a value of 1 makes RQ comparisons easier.
The following reference [27] was added to the end of section 2.6. Relative mRNA Expression by Real-time PCR.
[27] Livak, K.J.; Schmittgen, T.D. Analysis of relative gene expression data using real-time quantitative PCR and the 2(-Delta Delta C(T)) Method. Methods 2001, 25, 402-408.

Round 2
Reviewer 1 Report
the manuscript can be accepted in the present form